# Influence of Government Information on Farmers’ Participation in Rural Residential Environment Governance: Mediating Effect Analysis Based on Moderation

**DOI:** 10.3390/ijerph182312607

**Published:** 2021-11-30

**Authors:** Bowen Wang, Desheng Hu, Diandian Hao, Meng Li, Yanan Wang

**Affiliations:** College of Economics and Management, Northwest A&F University, Yangling 712100, China; hds_110@nwafu.edu.cn (D.H.); hdd197824@nwafu.edu.cn (D.H.); 15847108868@nwafu.edu.cn (M.L.); wyn3615@nwafu.edu.cn (Y.W.)

**Keywords:** rural residential environment governance, government information, depth of participation

## Abstract

Rural revitalisation in China relies heavily on the rural residential environment and is vital to the well-being of farmers. The governance of rural human settlements is a kind of public good. The external economy of governance results in the free-riding behaviour of some farmers, which does not entice farmers to participate in governance. However, current research seldom considers the public good of rural human settlements governance. This research is based on the pure public goods attribute of rural human settlements governance. It begins with government information and, using structural equation modelling (SEM), researchers construct the influence mechanism of government information, attitude, attention, and participation ability on the depth of farmers’ participation. The empirical results show that ability, attention, and attitude all have a dramatic positive influence on the depth of farmers’ participation, and the degree of impact gradually becomes weaker. Additionally, government information stimulus is not enough to promote farmers’ deep participation in governance. It needs to rely on intermediary variables to indirectly affect the depth of participation (ability, attention, attitude), and there is a path preference for the influence of government information on the depth of participation. As an important organisation in the management of rural areas, the village committee can significantly adjust the effect of the degree of attention on the depth of participation of farmers. Therefore, the government not only needs to provide farmers with reliable and useful information, but also needs to combine necessary measures to guide farmers to participate in the governance of rural human settlements.

## 1. Introduction

Improving upon and building a beautiful and liveable rural human settlement environment are important parts of the implementation of the rural revitalisation strategy in China. Due to the limitations of the urban-rural dual system, China ignored the construction of rural human settlements in the early stage of development. This led to the uneven development of urban and rural areas [1]. The problems of air pollution, garbage pollution, sewage pollution, and agricultural non-point source pollution in rural areas have been becoming increasingly serious [2,3,4]. The degree of pollution exceeds even that of large cities, which gradually threatens the happiness and health of farmers. As an important place for farmers’ the rural human settlement environment undertakes the important function of providing farmers with the necessary means of production and subsistence. However, the current rural human settlement environment is facing a series of problems, including serious pollution problems in rural areas [5], lack of effective management and protection of infrastructures [6], inadequate basic public services [7], and poor village appearances [8] etc. The issue of rural human settlements has gradually attracted the attention of the Chinese government. In 2018, the Chinese government proposed the “Three-year Action Programme for the Improvement of Rural Residential Environment,” and the nationwide rural living environment improvement action has started as well. The No. 1 Central Document in 2021 pointed out the implementation of the five-year action plan to refine the rural residential environment, focusing on the village toilet revolution, domestic garbage, and sewage treatment, while also strengthening the construction of rural public infrastructure and improving the level of rural basic public services.

The governance of rural human settlements not only included the governance of the rural environment but also involved rural infrastructure, social services, living conditions, environmental sanitation, etc. [5]. Paul Samuelson pointed out that the basic attributes of public goods include non-competitiveness and non-exclusiveness. The improvement of rural human settlements can not only improve the welfare of farmers themselves but must also benefit the people around them. Therefore, the rural human settlements environment is non-exclusive. According to Samuelson’ s theory, the rural residential environment conformed to the attributes of pure public goods [9]. As a public good, the government plays the main role in the governance of rural human settlements. However, relying solely on government governance, treating farmers as passive recipients, and ignoring the needs of farmers might cause problems such as low governance efficiency [10]. Additionally, the traditional top-down governance model, which is mainly based on theory and ignores public needs, could easily waste resources and create mismatches between supply and demand [11]. Public participation is an innovative way to solve public governance problems [12], which would not only reflect the public’ s real needs but also reduce the cost of government governance and improve governance efficiency. The governance of rural human settlements involves the utilisation and governance of rural public environment, resources, etc. The participation of farmers could effectively improve environmental quality [13]. In reality, information asymmetry restricts farmers’ participation in the governance of rural human settlements. As the most authoritative management organisation, it should be the responsibility of the Chinese government to help eliminate the impact of information asymmetry, as government information would guide farmers to take part in the governance of the rural living environment. Although the Chinese government actively communicates information about rural human settlement environment governance to farmers in a variety of ways, it does not have a significant impact on the participation of farmers or their behaviour toward governance. Thus, this paper will mainly answer the following questions. Can government information affect farmers’ deep participation in rural human settlements governance? What is the path that government information affects farmers’ deep participation in governance? Whether farmers’ deep participation in governance is affected by the external environment and what is the impact mode?

Based on the pure public goods attribute of rural human settlements governance, this paper aims to analyze how government information affects farmers’ deep participation in governance. This paper explores the influence mechanism of government information on farmers’ deep participation in governance from farmers’ factors and external factors. The results show that government information can not directly affect farmers’ deep participation in governance, it needs to rely on intermediary path, and external factors can significantly regulate farmers’ behavior. Therefore, our contributions are summarized as follows. Considering the information asymmetry in rural human settlements governance, this paper focuses on the impact of government information on farmers’ participation behavior. At the same time, the governance of rural human settlements involves many aspects and is a comprehensive governance project. Single participation in a certain link can not produce ideal results. Therefore, this paper focuses on the depth of farmers’ participation in governance. The above research results would make better use of the important role of government information in governance, which is conducive to enhance the ability of rural human settlement environment governance and builds an effective management and protection mechanism.

Based on the above analysis, this paper puts forward the following research hypotheses:

The rural human settlement environment is an important place for farmers to carry out production and life. Although farmers realise that they can improve their own welfare by improving the rural residential environment, because the rural residential environment has the attribute of public goods, there is information asymmetry. This severely hinders farmers’ participation in governance. Eliminating information asymmetry could effectively promote farmers’ willingness to participate [14]. Effective information content diffusion (regarding rural human settlement environment governance) could not only eliminate the negative impact of information asymmetry, but it could also guide individuals to take environmentally friendly behaviours [15]. Controlling the diffusion of negative information helps to promote these individual green behaviours [16]. As the direct organisation that manages the rural areas, the village committee acts as the main disseminator of government information and is the most dependent channel for farmers whose behaviour encourages them to seek information [17]. As shown in Figure 1, this study assumes that government information can help promote farmers’ participation in the governance of rural human settlements. Government information could affect every individual fairly, but in reality, individual environmental-related behavioural decisions appear to have significant differences [18]. Thus, this study assumes that the influence of government information on farmers’ behavioural decision-making has an intermediary path (see Figure 1).

**Hypothesis** **1** **(H1).** 
*Government information can significantly and directly promote farmers’ deep participation in rural human settlements governance.*


**Hypothesis** **2** **(H2).** 
*Government information affects farmers’ deep participation through several intermediary paths.*


Individuals can obtain direct value from information, so they pay selective attention to information based on their own needs [19]. Krupka et al. verify through experiments that when individual attention is drawn to norms, individual behaviour becomes more inclined to comply with norms, so government information can affect farmers’ attention [20]. As individuals continue to pay more attention to certain information, individuals receive more information about the governance of rural human settlements, such as governance significance and skills. This works to change individual attitudes and ability to participate. Therefore, the degree of attention can affect both the attitude of farmers and the ability of farmers to participate (L_1_, L_2_). In addition, farmers’ long-term attention behavior will stimulate farmers’ participation behavior. Thus, this study puts forward the following assumptions.

**Hypothesis** **3** **(H3).** 
*Government information can indirectly affect farmers’ deep participation behavior through farmers’ attention, and there is a remote intermediary path.*


Economics and social psychologists have believed that information played a huge role in the formation of attitudes [21], so environmental attitudes can directly affect farmers’ environmental protection behaviours [22]. The more positive the individual’s attitude towards the environment, the greater the cost that the individual is willing to bear when implementing pro-environmental behaviours [23]. Attitudes not only affect farmers’ participation ability (L_3_), but they also directly affect farmers’ participation behaviour.

**Hypothesis** **4** **(H4).** 
*Government information can indirectly affect farmers’ deep participation behavior through farmers’ participation attitude, and there is a remote intermediary path.*


Farmers can judge their ability to participate in the governance of rural human settlements from the information obtained from the outside and, at the same time, improve their ability to participate through information processing, utilisation, and learning. Therefore, government information can enhance the ability of farmers to participate. MOA theory has deemed that the greater the individual’ s ability to participate in an activity, the greater the probability of individual participation [4]. For farmers, when they have excess time, energy, and capital, they hope to improve their living environment.

**Hypothesis** **5** **(H5).** 
*Government information can indirectly affect farmers’ deep participation behavior through farmers’ participation ability.*


In China, as the most basic unit of rural management, the village committee is an effective channel for farmers to obtain government information. The village committee’s support for farmers’ participation in the governance of rural human settlements can greatly reduce the difficulty of their participation and encourage them to participate. Therefore, this study uses organisational support as a moderating variable to explore its moderating effect on the basic analysis framework of rural human settlement environments.

**Hypothesis** **6** **(H6).** 
*Organizational support can significantly regulate farmers’ deep participation behavior.*


## 2. Methods

### 2.1. Evaluation Methods

The structural equation model is divided into measurement model and structural model. The measurement model reflects the relationship between latent variables and observation variables, and the structural model defines the relationship between latent variables [24]. Latent variables cannot be observed and measured directly and must be measured by explicit variables. Therefore, structural equation model has unique advantages in solving the research with a large number of latent variables. In addition, the structural equation model can estimate the measurement error between variables in the measurement process, and use multiple indicators to test the effectiveness and reliability between observed variables and their latent variables [25]. In terms of action effect, structural equation model can judge the action effect between latent variables through path coefficient, so as to reveal the direct effect, indirect effect and total effect of latent variable A on latent variable B [26]. Therefore, structural equation model is widely used in Social Science [27,28], behavior research [29], education [30] and other fields. This research framework involves the calculation and test of intermediate effect and regulatory effect, and there are many latent variables. Therefore, structural equation model is selected as the main research method in this study.

The government information, attitude, ability, and attention involved in the research framework are all latent variables that cannot be directly and accurately observed. Therefore, several measurable variables need to be measured. This model has been divided into two parts: structural model and measurement model (as follows), which is mainly used to analyse the interdependence between a set of potential facets [31]. SEM has had a huge advantage in testing and analysing mediation effects [32]. Considering that this study measures the depth of participation of farmers by the number of farmers participating in governance projects, the depth of participation of farmers is an ordinal variable. If the ordinal variable is regarded as continuous and obeying normal distribution data, the maximum likelihood estimation method (ML) would cause parameter estimation deviation [33]. Muthén proposed to use a robust weighted least squares means and variance estimation method (WLSMV) to solve the problem of parameter deviation caused by ML estimation [34]. Therefore, this study uses Mplus 8.0 software to estimate the model using the WLSMV method.

Measurement Model: (1)$$x\;=\;\Lambda_{x}\xi \;+\;\sigma$$
(2)$$y\;=\;\Lambda_{y}\eta \;+\;\varepsilon$$

Structural Model:(3)η = Βη + Γξ + ι 

$\Lambda_{x}$ and $\Lambda_{y}$ represent the factor load matrix corresponding to exogenous latent variable and endogenous latent variable, respectively, in Formulas (1) and (2). *σ* and *ε* are the residuals corresponding to (1) and (2). *η* and *ξ* represent endogenous and exogenous latent variables respectively, and Β and Γ represent the effect coefficient matrix corresponding to endogenous and exogenous latent variables respectively in formulas (3). *ι* is the vector composed of the residuals of the structural model.

### 2.2. Measures of Latent Variables

Based on the theoretical framework, this study designs a complete set of measurement indicators around government information, attitudes, participation capabilities, attention, and organisational support. The questionnaire design adopts the Likert five-level scale method to obtain data. Points one through five indicate the degree from low to high (see Table 1 for specific measurement indicators and scale test results). For the measurement of the depth of participation, this study uses the main governance content proposed in the three-year action plan for the advancement of the rural residential environment in China (toilet revolution, domestic waste treatment, domestic sewage treatment, agricultural waste cleaning, infrastructure management, and protection). The depth of farmers’ participation in the rural human settlement environment is measured by the number of projects they participate in.

### 2.3. Data Sources

The data used in this paper comes from the household survey data done by the research team in Guanzhong area of Shaanxi Province in 2020. The survey is carried out in two stages. The first stage is carried out in August 2020, with a total of 1012 sample data recovered. The second stage is carried out in November 2020, with a total of 200 sample data recovered. In this survey, samples are selected according to the random stratified sampling method. Three regions of Xi’ an, Baoji and Xianyang are selected according to the economic gradient, and counties and towns are randomly selected in the three regions for investigation (Figure 2). According to the data in the statistical yearbook of Shaanxi Province in 2020, by the end of 2019, the proportion of secondary and tertiary industries in the sample area was 97.0%, 92.0% and 86.8% respectively, and the gradient of economic development level was obvious. All sample data involve 10 townships and 30 administrative villages. In order to ensure the quality of the questionnaire and avoid farmers’ misunderstanding of the questionnaire, this study used the form of one-to-one interview to obtain data, and distributed 1212 farmers’ samples, including 19.67% in Xi’an, 28.79% in Baoji and 51.54% in Xianyang. Combined with the needs of the research content, after eliminating outliers, missing values and nonparticipating samples, 976 samples were determined to be used in this study.

### 2.4. Data Analysis

Table 2 shows the basic characteristics of the survey sample and the depth of participation. The survey results indicate that in the survey samples, there are more women (56.66%) than men (43.34%). The aging of the survey samples is more obvious, and the age group is mainly over 60 years old (44.26%). In terms of education level, the proportion of junior middle school education is high, about 42.32%. Pure farmers accounted for 50.82% of the total sample, and only 1.64% of the survey sample were village officials. As for the depth of participation of farmers, 46.93% of the samples participated in two governance projects. With the increase of governance projects, the number of participating farmers gradually decreased.

## 3. Literature Analysis

Classical economics considers people to be rational subjects [35]. Thus, driven by the of profits, individuals’ behavioural decisions are often based on the principle of efficiency [36]. In respect to these ideas, then, it can be expected that farmers will want to modify their living conditions by participating in the governance of rural residential environment, thereby enhancing their own well-being. This then forms a motivation for farmers to participate. The Value-Belief-Norm theory (VBN), proposed by Stern [37], claims that ethics is the direct cause of behaviour and is related to individual values and beliefs. According to the VBN theory, individuals participating in environmental governance behaviours are driven by their own moral obligations [38]. The self-interest-oriented planning behaviour theory (TPB) guides individual behaviour through the judgment of risks and benefits [39] and tries to predict individual behaviour through individual intentions. TPB theory believes that individual behaviour is generated after the formation of willingness, and willingness is mainly affected by behaviour, attitude, subjective norms, and perceived behaviour control [40]. This theory has been widely used to predict various behaviours [41,42,43] and pays attention to individual behaviour motivation [44], ignoring individual participation opportunities and abilities. The Motivation–Opportunity–Ability model (MOA) constructs an individual behavioural decision analysis framework that includes motivation, opportunity, and ability [45]. Motivation was not the only factor in farmers’ behavioural decision-making [46]. This study, based on TPB theory and MOA theory, aims to explore the participation of farmers and their behaviour towards the governance of rural human settlements.

At present, scholars analyse the decision-making influence mechanism of farmers’ participation in the governance of rural human settlements around factors such as farmers’ characteristics, such as willingness, cognition. Compared to females, males were more inclined to engage in environmentally friendly behaviours [47]. Additionally, education level and awareness of environmental improvement affected young male farmers’ participation in the decision-making of rural human settlement environment governance [48]. TPB theory better explains the relationship between environmental behaviour and willingness of farmers [47], so it has been widely used in the research of farmers’ environmental behaviour and willingness. The willingness of farmers for ecological protection could significantly promote the behaviour of farmers for ecological protection [49,50], and willingness was a key factor that affected the actual environmental behaviour of farmers [51]. Hines et al. analysed 128 types of environmental behaviours and found that environmental cognition could significantly affect individual environmental behaviours [52]. In waste sorting and processing, although individuals with high cognition were more inclined to take participatory behaviours [53], the deviation between environmental cognition and behaviour [54] made cognition inaccurately predict individual behaviour. When using TPB theory to analyse farmers’ participation in behaviour, scholars found that farmers’ behaviours and attitudes could not only indirectly affect actual behaviour through willingness, but it also has a significant direct impact on actual behaviour [50]. Studies have found that attitudes play a mediating role in the influence of variables such as environmental awareness and environmental care on the governance of rural human settlements [55]. In the pro-environmental behaviours of farmers, the behavioural norms not only impact the behaviours of farmers, but behavioural norms also differ between age groups. For example, young farmers were more affected by personal norms, while older labourers were more affected by social norms [56]. Some scholars have combined motivation theory and TPB theory to verify that motivation significantly promotes farmers’ subjective norms, perceptual behaviour control, and other cognitive dimension factors, while cognitive dimension positively correlated with behavioural dimensions such as willingness and behaviour [51,57]. The above research is mainly based on the farmers’ own factors to explore the laws of rural human settlement environment governance. 

In addition, some scholars were concerned with different factors, including the behaviour of farmers in the same village [58], environmental governance policies [59], village organization [60], and other external factors. These factors prove to have significant effects on the behaviour of households’ rural human settlements. Wood et al. pointed out that farmers regard people around them as their main source of advice [61]. In agricultural environmental planning, the views of neighbouring farmers have a significant positive impact on farmers’ participation in decision-making [62]. In China, village committees, the most basic unit of rural management organisations, are of great significance to mobilise farmers’ willingness to participate and give play to their roles. A good relationship with other farmers could significantly increase a farmer’s willingness to participate in rural infrastructure maintenance [6].

At present, scholars have carried out more in-depth research on the environmental behaviour of rural households. The governance of rural human settlements is important environmentally, and the research foundation of the predecessors provides an important theoretical basis for this research. This study argues that there are still many shortcomings. Firstly, the governance of rural human settlements is a kind of public good, and the external economy of governance has caused some farmers to free ride, which reduces the farmers’ enthusiasm for participating in governance. However, current research seldom considers the attributes of public goods in the governance of rural human settlements. Secondly, existing research finds an inconsistency between farmers’ environmental behaviour and willingness [63], and it is difficult to accurately grasp the laws of farmers’ behaviour that focuses on their willingness to participate. The administration of rural residential environment involves environmental governance, infrastructure construction, public services, toilet revolution, garbage disposal, sewage treatment, etc. It is difficult to boost the overall improvement of rural residential environment governance by simply studying the participation of farmers in a certain link. Therefore, it is more important to explore the impact mechanism of the depth of farmers’ participation. Thirdly, in the current research on the environmental behaviour of farmers, scholars tend to study either factors of the farmers or external factors alone. In reality, the behavioural decisions of farmers are affected by both their own factors and external factors. For the rural residential environment with pure public goods attributes, the cooperative governance model is more efficient than single-subject governance [64]. It is necessary to consider the influence of government variables on the rural residential environment governance behaviour of farmers.

## 4. Results

### 4.1. Reliability and Validity of the Scale

According to the theoretical framework, this study obtains relevant date through questionnaires, then designs a five-level scale around the five aspects of government information, attitude, ability, attention, and organisational support. This paper uses SPSS 26 (Inc., Chicago, IL, USA) and Mplus 8.0(Muthén & Muthén, Los Angeles, CA, USA) to test the reliability and validity of the scale. As presented in Table 1, the overall KMO value of the scale is 0.839, which is greater than the minimum standard value of 0.5. Bartlett’s sphericity test significantly rejects the null hypothesis, indicating that the scale used in this article is suitable for factor analysis. The standardised factor loading, Cronbach’ s α, C.R., AVE of each latent variable were greater than 0.6, 0.7, 0.7, and 0.5, respectively. In summary, the scale used in this article has good reliability and validity.

### 4.2. Analysis on Influencing Factors of Farmers’ Participation Depth

#### 4.2.1. Fitting of Model

If the model cannot fit the data well, it will be difficult to use the model to explain real problems well. In this study, Mplus 8.0 software was used to estimate the model using the WLSMV method, and the model in the article was tested and analysed with reference to the analysis steps and standards of William [65]. As shown in Table 3, the key indexes for evaluating the fitting degree of SEM are up to the standard, indicating that the model constructed in this study has a good fitness.

#### 4.2.2. Interpretations of the Estimation Results

Figure 3 displays the relationship of interaction between latent variables. As shown in Table 4, different latent variables each have different degrees of effect on the depth of farmers’ participation. Firstly, government information has a significant positive impact on farmers’ attitudes, participation ability, and attention to rural human settlements at the significance level of 0.001. However, government information cannot significantly affect the depth of farmers’ participation. Secondly, the attitude of farmers can positively improve the ability of farmers to participate, with a significance level of 0.001, which shows that a positive attitude towards participation will encourage farmers to invest more time, energy, and capital in the governance of rural human settlements. At the 0.05 level of significance, the attitude of farmers is conducive to guiding farmers to engage more deeply in the governance of the rural residential environment. Thirdly, the ability to participate significantly positively affects the depth of participation of farmers at the level of 0.001. Fourthly, the degree of attention has a significant role in promoting attitude, ability, and depth of participation. It is worth noting that there are differences in the degree of influence of different latent variables on the depth of farmers’ participation. The standardised coefficient of the influence of participation ability on the depth of participation of farmers is the highest (β = 0.210; *p* < 0.001), followed by the degree of attention (β = 0.106; *p* < 0.01). Therefore, in practice, the focus should be on guiding farmers to participate in governance by improving their participation ability and attention. The participation attitude of farmers only affects the depth of participation at the 0.05 significance level, and the impact is the weakest (β = 0.077).

#### 4.2.3. Analysis of Mediating Effect

This study uses the bootstrapping method to test the mediating effect of government information affecting the depth of farmers’ participation. Table 5 indicates the influence mechanism of government information on the depth of farmers’ participation. The research results demonstrate that government information mainly affects the depth of farmers’ participation through different intermediary paths, while the direct effect of government information is not significant. In terms of the mediating effect, government information can significantly affect the depth of participation of farmers through participation ability (0.026) and attention (0.027), and the mediating effect of attitude is not significant. For remote intermediaries, government information can indirectly affect the depth of participation through attitude and ability (0.009) and can affect the depth of participation through attention and ability (0.016). Although the intermediary path of government information, attention, attitude, participation, and depth is not significant, after the introduction of participation ability, this remote intermediary becomes significant (β = 0.003; *p* < 0.05). Government information can affect the depth of participation through different intermediary paths. As to whether there are significant differences between different paths, further analysis is needed. Hence, this study still uses the bootstrapping method to test the differences between the paths. Considering the space limitation, this study only lists the paths with significant differences. As shown in Table 5, there are significant differences between the R_3_ path and the R_4_, R_6_, and R_7_ paths, and the effects of the R_3_ path are stronger than the effects of the three.

#### 4.2.4. The Moderating Effect of Organisational Support

Based on the previous theoretical framework, this study introduces organisational support as the moderator of the model and assumes that organisational support adjusts the relationship between the ability to participate in the participation depth and the attention to the participation depth. Therefore, this study uses the Monte Carlo method to analyse the moderating effect of organisational support. However, using Monte Carlo analysis needs to determine the minimum sample size to ensure the statistical power and accuracy of parameter estimation. Line (2005) and Raykov (2006) gave suggestions on sample size. The absolute sample size is not less than 200, and the ratio of sample size to the number of variables is greater than 10 [66,67]. The estimation of the minimum sample size is affected by the type, quantity and distribution of observed variables. According to the standards of Muthén and Muthén [68], 976 samples used in this study can be estimated more accurately.

Table 6 exhibits that organisational support does not significantly regulate the effect of ability on the depth of participation, but organisational support affects the attention on the depth of participation (0.112) at a significance level of 0.05, so as the degree of organisational support increases, the degree of attention has a stronger effect on the depth of participation.

## 5. Discussion

Based on the perspective of farmers, this study combines the TPB and MOA theories to construct a theoretical analysis framework of government information, attitudes, participation ability, attention, and depth of participation. This study uses the survey data of farmers in the Guanzhong area of Shaanxi Province and uses SEM to verify the theoretical framework.

The governance of rural human settlements is a kind of public good, and the reason for the free-riding behaviour of some farmers is the asymmetry of information among farmers and the lack of consensus on action. As the most authoritative and credible institution in the management of rural areas, the Chinese government can use the power of administrative management to deliver effective information to farmers. This would help to eliminate the problem of information asymmetry and would guide farmers to form a governance consensus. The four measurable variables used in this study to measure government information (information accuracy, information clarity, information comprehensive and detail level, and information timeliness) are all good measures of government information. If the information provided by the government to farmers meets these four requirements, it can effectively eliminate the problem of information asymmetry, reduce farmers’ free-riding behaviour, and encourage farmers to participate in the governance of rural human settlements.

In reality, the channels through which farmers receive government information are the equal (village committees, the Internet, etc.), and government information has a fair effect on farmers. The empirical results state that government information cannot significantly and directly affect the depth of farmers’ participation. However, in reality, there are differences in the depth of farmers’ participation. This difference is not irrelevant to government information. Government information can affect farmers’ attitudes and attention towards the administration of rural residential environments, and farmers’ attitudes and attention will affect their ability to participate. The differences in farmers’ attitudes, attention, and ability to participate in the governance of rural human settlements have caused differences in the depth of farmers’ participation. Therefore, the impact of government information on the depth of farmers’ participation requires the use of intermediary paths, such as attitude, attention, and ability to participate. The research results verify this. Therefore, relying solely on government information is difficult to effectively guide farmers to participate in the governance of rural human settlements. Effective governance measures need to combine government information with farmers’ attitudes, attention, and participation capabilities in order to urge farmers to take part in the governance of rural human settlements.

The research results show that the degree of influence of participation ability, attention, and attitude on the depth of participation is gradually weakening. Participating in the governance of rural human settlements requires time, manpower, and capital, and these objective resource inputs have a huge restrictive effect on farmers’ participation in governance. Therefore, the key to affecting farmers’ participation in the governance of rural human settlements is to improve their ability to participate. Participation capacity, as an effective intermediary of government information for the depth of participation, can significantly promote farmers’ in-depth participation in the governance of rural human settlements. Attention, as another intermediary variable of government information and participation depth, cannot only increase the depth of farmers’ participation by strengthening participation ability but also promote farmers’ in-depth participation by influencing their attitudes and strengthening participation ability. Continuous attention to the governance of rural human settlements has enabled farmers to better understand the meaning of governance and continuously improve their attitudes. Moreover, their continuous attention behaviour helps farmers to quickly obtain the latest information and take timely actions. It is worth noting that the intervention of the organisation can effectively improve the effect of farmers’ attention on the depth of participation, which provides a direction for the organisation to guide farmers to participate in governance. Compared with participation ability and attention, attitude reflects more of the psychological state of farmers, whereas other factors are needed to stimulate the production from psychological state to actual behaviour. Therefore, attitude has the weakest influence on the depth of participation and is not an effective mediating variable. Through the results of path difference analysis, this study found that there are significant differences in the effects of certain paths, such as R_3_, R_4_; R_3_, R_6_, etc., indicating that there is a path preference for the influence of government information on the depth of participation.

## Figures and Tables

**Figure 1 ijerph-18-12607-f001:**
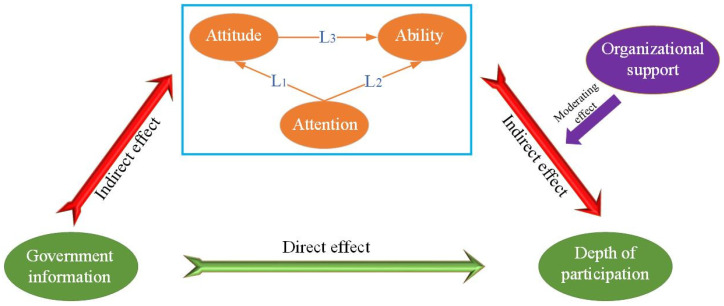
The theoretical framework of farmer participation depth.

**Figure 2 ijerph-18-12607-f002:**
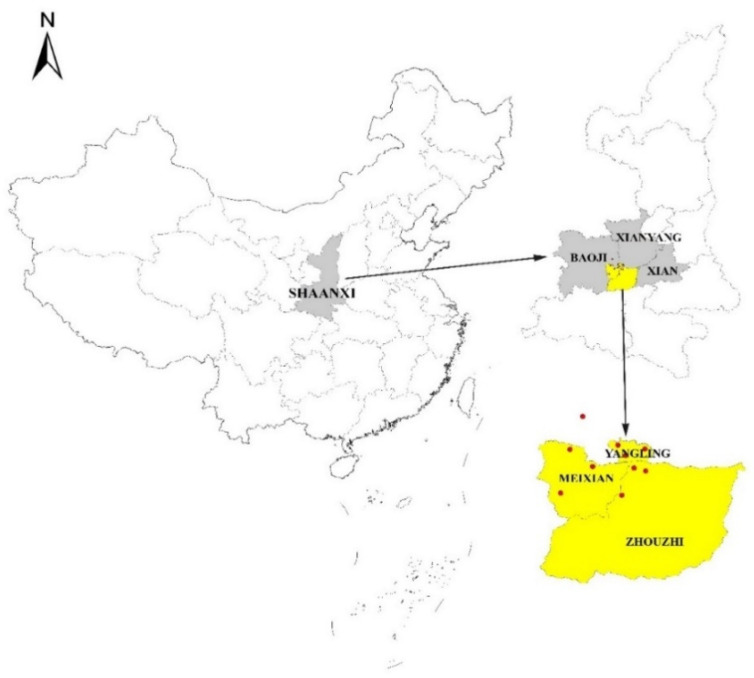
Study area.

**Figure 3 ijerph-18-12607-f003:**
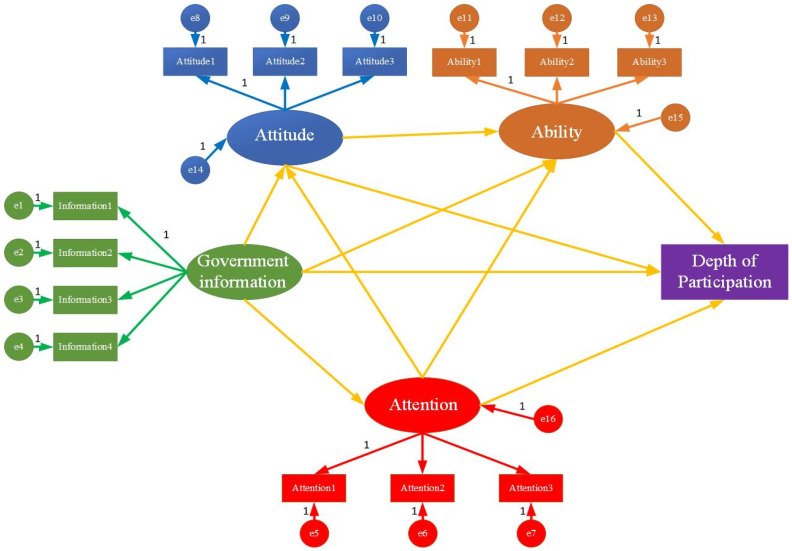
Structural equation model.

**Table 1 ijerph-18-12607-t001:** Reliability and validity test.

Variable		Variable Interpretation	Mean	Std. Dev	Std. F.L.	Cronbach’ s α	C.R.	AVE
Govern-ment informa-tion (F1)	Infor_1_	Accurate information of rural human settlements	3.76	0.694	0.840	0.916	0.919	0.740
	Infor_2_	Clear information of rural human settlements	3.75	0.707	0.924			
	Infor_3_	Full and detailed information of rural human settlements	3.76	0.752	0.905			
	Infor_4_	Timeliness of information of rural human settlements	3.65	0.766	0.762			
Attitude(F2)	Atti_1_	Conductive to village planning	4.09	0.676	0.915	0.871	0.883	0.718
	Atti_2_	Improve the living environment	4.12	0.679	0.883			
	Atti_3_	Get approval from others	3.84	0.812	0.733			
Ability (F3)	Abi_1_	Bear the cost	3.20	0.963	0.636	0.750	0.755	0.508
	Abi_2_	Have time to participate	3.33	0.962	0.761			
	Abi_3_	Ability to participate	3.56	0.954	0.735			
Attenti-on(F4)	Attet_1_	Often follow	3.69	0.836	0.834	0.837	0.839	0.634
	Attet_2_	Actively share relevant content	3.54	0.855	0.741			
	Attet_3_	Continue to follow in the future	3.71	0.801	0.812			
Organis-ational Support (F5)	OS_1_	Village committee encourages participation	3.59	0.878	0.854	0.876	0.878	0.707
	OS_2_	Village committees create opportunities for participation	3.63	0.857	0.902			
	OS_3_	Village committee values	3.42	0.871	0.760			

KMO = 0.839 Bartlett = 8904.395 DF = 120 Sig. = 0.000

**Table 2 ijerph-18-12607-t002:** Sample description.

Survey Targets		Sample Size	Percentage (%)	Survey Targets		Sample Size	Percentage (%)
Sex	Female	553	56.66		>60	432	44.26
	Male	423	43.34	Education	Illiterate	138	14.14
Participation depth	1	83	8.50		Primary	225	23.05
	2	458	46.93		Middle	413	42.32
	3	244	25.00		High or vocational	130	13.32
	4	108	11.07		College and above	70	7.17
	5	83	8.50	Identity	Village cadres	16	1.64
Age	≤30	143	14.65		Ordinary villagers	960	98.36
	31–40	77	7.89	Pure farmer	1	496	50.82
	41–50	90	9.22		0	480	49.18
	51–60	234	23.98				

**Table 3 ijerph-18-12607-t003:** Model fitting.

Statistical Test	Standard Values of Fit Index	Actual Fitting Results	Test Results
χ^2^/df values	Between 1 and 3	2.872	Good fit
RMSEA	<0.05	0.044	Good fit
CFI	>0.9	0.957	Good fit
TLI	>0.9	0.943	Good fit
SRMR	<0.08	0.024	Good fit

**Table 4 ijerph-18-12607-t004:** Structural model standardisation coefficient.

Variable Relationship	Estimate	S.E.
F1 to F2	0.151 ***	0.029
F1 to F3	0.124 ***	0.031
F1 to F4	0.252 ***	0.029
F1 to participation depth	0.063	0.035
F2 to participation depth	0.077 *	0.039
F2 to F3	0.287 ***	0.033
F3 to participation depth	0.210 ***	0.044
F4 to F2	0.206 ***	0.027
F4 to F3	0.310 ***	0.034
F4 to participation depth	0.106 **	0.037

*, **, *** was significant at 0.05, 0.01 and 0.001 levels, respectively.

**Table 5 ijerph-18-12607-t005:** Mediation effect and difference test.

	Estimate	Product of Coefficients		Bootstrapping			
				Percentile 95% CI		BC 95% CI	
		S.E.	Z	Lower	Upper	Lower	Upper
Total	0.160 ***	0.039	4.078	0.084	0.238	0.083	0.238
Total Indirect	0.097 ***	0.019	4.973	0.063	0.138	0.063	0.138
Direct	0.063	0.045	1.398	−0.024	0.150	−0.024	0.149
F1 to F2 to participation depth (R1)	0.012	0.009	1.315	−0.003	0.031	−0.001	0.035
F1 to F3 to participation depth (R2)	0.026 *	0.012	2.128	0.006	0.055	0.007	0.056
F1 to F4 to participation depth (R3)	0.027 *	0.013	2.011	0.003	0.055	0.004	0.057
F1 to F4 to F2 to participation depth (R4)	0.004	0.003	1.533	−0.001	0.010	0.000	0.011
F1 to F2 to F3 to participation depth (R5)	0.009 *	0.004	2.263	0.003	0.019	0.003	0.021
F1 to F4 to F3 to participation depth (R6)	0.016 *	0.006	2.53	0.006	0.031	0.007	0.032
F1 to F4 to F2 to F3 to participation depth(R7)	0.003 *	0.002	1.994	0.001	0.007	0.001	0.008
Contrasts							
R_4_ vs. R_3_	−0.046 **	0.018	−2.627	−0.087	−0.018	−0.090	−0.019
R_6_ vs. R_3_	−0.045 **	0.017	−2.598	−0.087	−0.018	−0.089	−0.019
R_7_ vs. R_3_	−0.048 **	0.017	−2.780	−0.087	−0.019	−0.092	−0.021

Note: *, **, *** was significant at 0.05, 0.01 and 0.001 levels, respectively; BC, bias corrected; 5000 bootstrap samples.

**Table 6 ijerph-18-12607-t006:** Moderating effect test.

Variable Relationship	Estimate	S.E.
F1 to F2	0.151 ***	0.029
F1 to F3	0.124 ***	0.031
F1 to F4	0.252 ***	0.029
F1 to participation depth	0.063	0.035
F2 to participation depth	0.077 *	0.039
F2 to F3	0.287 ***	0.033
F3 to participation depth	0.210 ***	0.044
F4 to F2	0.206 ***	0.027
F4 to F3	0.310 ***	0.034
F4 to participation depth	0.106 **	0.037
F5 to participation depth	0.049	0.049
F5 * F3 to participation depth	−0.073	0.050
F5 * F4 to participation depth	0.112 *	0.049

Note: *, **, *** was significant at 0.05, 0.01 and 0.001 levels, respectively.

## Data Availability

The datasets used and analyzed during the current study are available from the correspondent author on reasonable request.

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
