# Peer review of "Influence of Government Information on Farmers’ Participation in Rural Residential Environment Governance: Mediating Effect Analysis Based on Moderation"

_ijerph, 2021, doi:10.3390/ijerph182312607_

Round 1

Reviewer 1 Report

  1. Authors are advised to describe the sampling method in more details.
  2. Authors are recommended to provide the rational for the sample size used in this study; if possible, Monte Carlo simulation is suggested to perform to determine the appropriate sample size.
  3. It would add to the completeness of the paper if demographic variables of farm households are also examined.
  4. To my personal point of view, I will argue that it would be more appropriate to look into the willingness to pay of farmers in terms of their participating in rural residential settlement; nonetheless, each approach has its own pro and con, as long as it is built upon a sound theoretical basis; for the estimation method, of course, it has several methods for estimating SEM model when dealing with ordinal variables, yet each one has its own advantages as well as disadvantages. To sum it up, I think you did a good job for this paper.

References:

Wolf, E. J., Harrington, K. M., Clark, S. L., & Miller, M. W. (2013). Sample Size Requirements for Structural Equation Models: An Evaluation of Power, Bias, and Solution Propriety. Educational and psychological measurement76(6), 913–934. https://doi.org/10.1177/0013164413495237 

Wang, W., H. Gong, L. Yao, & L. Yu. (2021). ,Preference Heterogeneity and Payment Willingness within Rural Households’ Participation in Rural Human Settlement Improvement. Journal of Cleaner Production, 312, 127529.

Author Response

Dear Reviewer 1:

Thank you for the reviewer’s comments concerning our manuscript entitled “Influence of government information on farmers' participation in rural residential environment governance: Mediating effect analysis based on moderation” (Manuscript ID: ijerph-1433308). Those comments are valuable and very helpful for revising and improving our paper, as well as the important guiding significance to our researches. We have studied comments carefully and have made correction which we hope meet with approval. The main corrections in the paper and the response to the reviewer’s comments are as following:

Point 1: Authors are advised to describe the sampling method in more details.

Response 1: We are very grateful for your valuable instructive advice. We supplemented the details of sampling methods and the research process in data sources. The current data sources are as follows:

“The data used in this paper comes from the household survey data done by the research team in Guanzhong area of Shaanxi Province in 2020. The survey is carried out in two stages. The first stage is carried out in August 2020, with a total of 1012 sample data recovered. The second stage is carried out in November 2020, with a total of 200 sample data recovered. In this survey, samples are selected according to the random stratified sampling method. Three regions of Xi'an, Baoji and Xianyang are selected according to the economic gradient, and counties and towns are randomly selected in the three regions for investigation (Fig. 2). According to the data in the statistical yearbook of Shaanxi Province in 2020, by the end of 2019, the proportion of secondary and tertiary industries in the sample area was 97.0%, 92.0% and 86.8% respectively, and the gradient of economic development level was obvious. All sample data involve 10 townships and 30 administrative villages. In order to ensure the quality of the questionnaire and avoid farmers' misunderstanding of the questionnaire, this study used the form of one-to-one interview to obtain data, and distributed 1212 farmers' samples, including 19.67% in Xi'an, 28.79% in Baoji and 51.54% in Xianyang. Combined with the needs of the research content, after eliminating outliers, missing values and non-participating samples, 976 samples were determined to be used in this study.”

Point 2:Authors are recommended to provide the rational for the sample size used in this study; if possible, Monte Carlo simulation is suggested to perform to determine the appropriate sample size.

Response 2: Thank you for your thoughtful comment. We refer to the previous methods of calculating the sample size through Monte Carlo, and try to conduct Monte Carlo simulation through Mplus to calculate the reasonable sample size value, so as to make the accuracy and efficiency of the estimated parameters at an acceptable level. Although we refer to mplus8.0 user manual to write Monte Carlo simulation commands, due to the use of interactive variables in this study, we have not been able to write concise and effective commands to calculate reasonable sample size values. Fortunately, we found some reliable achievements in the study of Monte Carlo simulation to prove that the sample size used in this paper is reasonable, and described it in detail in this paper. As follows:

“Based on the previous theoretical framework, this study introduces organisational support as the moderator of the model and assumes that organisational support adjusts the relationship between the ability to participate in the participation depth and the attention to the participation depth. Therefore, this study uses the Monte Carlo method to analyse the moderating effect of organisational support. However, using Monte Carlo analysis needs to determine the minimum sample size to ensure the statistical power and accuracy of parameter estimation. Line (2005) and Raykov (2006) gave suggestions on sample size. The absolute sample size is not less than 200, and the ratio of sample size to the number of variables is greater than 10 [66,67]. The estimation of the minimum sample size is affected by the type, quantity and distribution of observed variables. According to the standards of Muthén and Muthén [68], 976 samples used in this study can be estimated more accurately.”

Point 3: It would add to the completeness of the paper if demographic variables of farm households are also examined.

Response 3: Thank you for your valuable comment. We checked the demographic variables by reviewing the data. In order to increase the integrity of the paper, we added the mean and variance of variables in Table 2 of the first draft, and marked them red in the revised draft.

Point 4: To my personal point of view, I will argue that it would be more appropriate to look into the willingness to pay of farmers in terms of their participating in rural residential settlement; nonetheless, each approach has its own pro and con, as long as it is built upon a sound theoretical basis; for the estimation method, of course, it has several methods for estimating SEM model when dealing with ordinal variables, yet each one has its own advantages as well as disadvantages. To sum it up, I think you did a good job for this paper.

Response 4: Thank you very much for your advice. We also believe that it is very valuable to study the willingness to pay of farmers to participate in rural human settlements governance. In the future research, we plan to further study the research on farmers' willingness to pay. In addition, SEM model is a very rich and interesting model. Through continuous learning, we hope to apply SEM model to more research.

We tried our best to modify this paper and undergo several revisions to improve the manuscript. These changes will not influence the content and framework of the paper. And here we did not list the changes but marked in red in revised paper. We appreciate for Reviewer 1’s warm work earnestly, and hope that the correction will meet with approval.

Once again, thank you very much for your comments and suggestions.

Reviewer 2 Report

The topic of the article is very interesting, however, some corrections should be made in order to make it suitable for publication
1) The structure of the text should include: introduction, methods, literature analysis, results, discussion and citation. 
2) The introduction should pose research questions or theses/hypotheses - this is missing here
3) The chosen methods should be justified and the materials used for the research should be identified
4) The methodological aspect should be justified
5) It is not clear why the method of structural equations was chosen (the framework presented by the authors is not well described, there is no reference to the theory in this area, the authors must justify the choice or indicate the innovativeness, novelty of their considerations)

Author Response

Point 1: The structure of the text should include: introduction, methods, literature analysis, results, discussion and citation.

Response 1: Thank you for your advice. In the revised manuscript, we readjusted the article structure. The revised article structure is as follows:

  1. Introduction
  2. Methods

2.1 Evaluation methods

2.2 Measures of latent variables

2.3 Data sources

2.4 Data analysis

  1. Literature analysis
  2. Results

4.1 Reliability and validity of the scale

4.2 Analysis on influencing factors of farmers' participation depth

4.2.1 Fitting of model

4.2.2 Interpretations of the estimation results

4.2.3 Analysis of mediation effect

4.2.4. The moderating effect of organisational support

  1. Discussion

References

Point 2: The introduction should pose research questions or theses/hypotheses, this is missing here.

Response 2: Thank you for your thoughtful comment. In the introduction, we elaborate the problems of the article in more detail, so that readers can quickly understand the content of the article, and we also summarize the research value of the article. In addition, based on the research problems and theoretical analysis, we put forward the research hypothesis and elaborate it in detail in the introduction. The revised content has been marked red in the revised text.

Point 3: The chosen methods should be justified and the materials used for the research should be identified.

Response 3: Thank you for your advice. Firstly, the theoretical framework proposed in this paper involves multiple latent variables, and structural equation model has unique advantages in studying the relationship between latent variables; secondly, the structural equation model estimates the measurement error between variables in the measurement process, and has mature and reliable test methods for the reliability, validity and model fitting of measurable variables; finally, we believe that the impact of government information on the depth of farmers' participation needs to be affected through several intermediary variables. Therefore, in order to accurately estimate the direct effect, indirect effect and total effect between latent variables, this paper selects structural equation model for analysis, and uses bootstrap to test the effect. In order to increase the reliability of the methods used in this paper, we described them in detail in "evaluation methods". As follow:

“The structural equation model is divided into measurement model and structural model. The measurement model reflects the relationship between latent variables and observation variables, and the structural model defines the relationship between latent variables[54]. Latent variables cannot be observed and measured directly and must be measured by explicit variables. Therefore, structural equation model has unique advantages in solving the research with a large number of latent variables. In addition, the structural equation model can estimate the measurement error between variables in the measurement process, and use multiple indicators to test the effectiveness and reliability between observed variables and their latent variables[56]. In terms of action effect, structural equation model can judge the action effect between latent variables through path coefficient, so as to reveal the direct effect, indirect effect and total effect of latent variable A on latent variable B[55]. Therefore, structural equation model is widely used in Social Science [57; 58], behavior research [59], education [60] and other fields. This research framework involves the calculation and test of intermediate effect and regulatory effect, and there are many latent variables. Therefore, structural equation model is selected as the main research method in this study.

Point 4: The methodological aspect should be justified.

Response 4: Thank you for your advice. Before using the SEM model, we performed EFA on the observed variables, the standardized factor load was more than 0.6, and the C.R. value and AVE value reached the standard. The model fitness test results show that the model constructed in this paper can fit the data better, and the method used in the text is reasonable.

Point 5: It is not clear why the method of structural equations was chosen (the framework presented by the authors is not well described, there is no reference to the theory in this area, the authors must justify the choice or indicate the innovativeness, novelty of their considerations).

Response 5: Thank you for your advice. After combing the literature on SEM model, we explained the reasons for using SEM model in detail in "evaluation methods" from the advantages of SEM model in dealing with latent variable relationship, research content, model test and so on. The specific modifications have been marked in red in the text.

Round 2

Reviewer 2 Report

The authors have addressed the comments and posted a revised version.